# Detection of A2143G, A2142C, and A2142G Point Mutations with Real-Time PCR in Stool Specimens from Children Infected with *Helicobacter pylori*

**DOI:** 10.3390/diagnostics12092119

**Published:** 2022-08-31

**Authors:** Nesrin Gareayaghi, Bekir Kocazeybek

**Affiliations:** 1Istanbul Sisli Hamidiye Etfal Training and Research Hospital, Center for Blood, University of Health Sciences, Istanbul 34098, Turkey; 2Department of Medical Microbiology, Cerrahpasa Medical Faculty, Istanbul University-Cerrahpasa, Istanbul 34098, Turkey

**Keywords:** *Helicobacter pylori*, clarithromycin resistance, symptomatic children, real-time PCR, point mutation

## Abstract

Reports have indicated an increasing prevalence of clarithromycin resistance in children relative to adults. Thus, it is important to investigate primary clarithromycin resistance before therapy to avoid treatment failure. A2142G, A2143G, and A2142C point mutations in the peptidyltransferase region of the 23S ribosomal RNA (rRNA) of *Helicobacter pylori* (*H. pylori*) strains isolated from children with gastrointestinal symptoms and asymptomatic children were evaluated via real-time polymerase chain reaction (RT-PCR) using fecal DNA samples. The presence of *H. pylori* was determined using a fecal *H. pylori* antigen enzyme-linked immunosorbent assay (ELISA) kit from the stools of children (*n* = 543). A2143G, A2142C, and A2142G point mutations were detected via RT-PCR and confirmed by sequencing the 23S rDNA. Fecal *H. pylori* antigen testing was positive in 101 symptomatic (49) and asymptomatic (52) children. A significant difference was found between the 0–5- and 5–18-year-old groups in terms of the A2143G and A2142G point mutations (*p* = 0.001). The A2142C mutation was not detected. There was a significant difference in the A2143G mutation between the symptomatic and asymptomatic 5–18-year-old children (*p* = 0.019). Macrolides are frequently used to treat upper respiratory tract infections in children due to their selective pressure effect. We suggest that *H. pylori* strains carrying mutations in the 23S RNA subunit conferring clarithromycin resistance may lead to an intense inflammatory response in the gastric epithelial cells, allowing them to proliferate more rapidly and causing possible diarrhea, halitosis, or abdominal pain in children.

## 1. Introduction

*Helicobacter pylori* (*H. pylori*) may cause chronic gastritis, peptic ulcers, gastric cancer, and mucosa-associated lymphoid tissue (MALT) lymphoma by colonizing the gastric mucosa; it is usually acquired in childhood [1]. In Turkey, 30–56.6% of children are thought to have *H. pylori* infections, and detecting and treating these children is an important health priority [2]. The detection of an *H. pylori* infection is generally achieved via the histological detection and culturing of the pathogen or by an invasive method such as serology, the ^13^C urea breath test, fecal *H. pylori* antigen tests, or DNA detection [3]. However, some serological tests are not used in children because they have low sensitivities. Fecal *H. pylori* antigen tests (enzyme immunoassays) based on monoclonal antibodies have been very promising in children, but no information related to the susceptibility or resistance of *H. pylori* to antibiotics has been obtained [3].

Information in the literature related to the antimicrobial susceptibility of *H. pylori* in children is limited because endoscopy is rarely indicated; therefore, culture and antimicrobial susceptibility tests are rarely performed [4]. According to the Maastricht IV Consensus, idiopathic thrombocytopenic purpura and iron deficiency anemia are extragastric disorders for which *H. pylori* infection detection and eradication are indicated, while diagnostic tests are recommended in pediatric patients with peptic ulcer disease in the guidelines of both the European Society for Paediatric Gastroenterology, Hepatology, and Nutrition (ESPGHAN) and the North American Society for Paediatric Gastroenterology, Hepatology, and Nutrition (NASPGHAN) [5,6]. The initial diagnosis of an active *H. pylori* infection in children with idiopathic thrombocytopenic purpura, iron deficiency anemia, and peptic ulcer disease should be based on either positive histopathology plus a positive rapid urease test or a positive culture as recommended by the guidelines from ESPGHAN/ NASPGHAN. If the histopathological examination and rapid urease test results do not match, it is necessary to carry out additional non-invasive tests such as the urea breath test (UBT) or the *H. pylori* stool antigen (PSA) test [7]. Invasive methods are not preferred in epidemiological studies of healthy children because endoscopic procedures are needed [8]. Reports have indicated an increased prevalence of clarithromycin resistance in children relative to adults [9]. Thus, it is important to investigate primary clarithromycin resistance before therapy.

Kato and Fujimura [10] reported a 36% clarithromycin resistance rate in children, and the rate was found to be 14.2–78% in another study [9]. The clarithromycin resistance rates of *H. pylori* strains in children infected with *H. pylori* vary between 13.9% and 84.9% in 10 countries and have increased over time [11]. Increases in clarithromycin resistance rates in children may be due to the excessive use of clarithromycin for the treatment of respiratory tract infections. Clarithromycin resistance in *H. pylori* can cause treatment failure in triple therapy (proton pump inhibitor in combination with two antibiotics, including amoxicillin, clarithromycin, or metronidazole) [3,4].

The detection of a gene mutation from fecal specimens is a noninvasive method that is easier and more cost-effective than traditional culture and susceptibility test techniques [12]. It is known that clarithromycin resistance in *H. pylori* strains is caused by three point mutations in the peptidyltransferase region of the 23S rRNA (A2142G, A2143G, and A2142C) [13]. The detection of these point mutations usually requires a cultured *H. pylori* strain or gastric biopsy specimen, which is obtained using several molecular methods [14]. These methods include polymerase chain reaction (PCR)-based sequencing analysis and restriction fragment length polymorphism (RFLP)-nested PCR [14]. However, new methods of detecting these mutations in human feces eliminates the need for endoscopy procedures while maintaining high sensitivity and specificity [15]. Cytotoxin-associated antigen A (cagA)-positive *H. pylori* strains are easily eradicated through the application of antibiotics due to their fast growth rate when compared with cag-A negative strains [14].

Here, A2142G, A2143G, and A2142C point mutations in the peptidyltransferase region of the 23S rRNA of *H. pylori* strains isolated from children with gastrointestinal symptoms and asymptomatic children were evaluated via the real-time polymerase chain reaction (RT-PCR) method using fecal DNA samples, and the association of cagA positivity with these point mutations was measured.

## 2. Materials and Methods

### 2.1. Study Design

This is a case–control study conducted between April 2019 and March 2021. A total of 543 children were included in the study (M:302, F:241). The mean age was 34.0 ± 34.1 months. The first 3 children who applied to the pediatric outpatient clinic on working days were asked to give stool samples regardless of their clinical characteristics. Children who had used antibiotics or proton pump inhibitors in the two weeks prior to the study, those aged over 18 years, and those who had undergone previous gastric surgery were excluded. Children with abdominal pain, halitosis, and diarrhea were regarded as symptomatic. The parents of the subjects were later asked to complete a questionnaire by telephone. The children were later stratified by age: 0–5 years and 5–18 years.

### 2.2. Determining the Positivity of H. pylori by Fecal H. pylori Antigen ELISA Kit

Stool samples were stored at −80 °C degrees until analysis. The presence of *H. pylori* was determined using a fecal *H. pylori* antigen ELISA kit (Epitope Diagnostics, Inc., CA, USA) while using urease-specific monoclonal antibodies as positive or negative controls. This test detects *H. pylori* antigen using monoclonal antibodies, and this test does not detect urease antigen.

### 2.3. Amplification of the H. pylori cagA Gene

The *cagA* gene was determined using a molecular PCR technique with specific primers [16]. All primer sets are shown in Table 1 [17,18]. The study protocol was as follows: initial denaturation at 95 °C for 2 min, followed by 45 cycles of 95 °C for 30 s, 45 s at 53 °C, and 45 s at 72 °C. The final elongation was performed for 5 min at 72 °C. All of the PCRs were conducted with a negative control (nuclease-free water) and a *cagA* positive control (*H. pylori* 26,695 strain).

### 2.4. Detection of A2143G, A2142C, and A2142G Point Mutations by Real-Time PCR

*H. pylori* DNA isolation from stool samples was performed using the High Pure PCR template preparation kit (Roche Diagnostics Mannheim, Mannheim, Germany) in accordance with the manufacturer’s instructions. The hybridization probes and primers in Table 1 were used for the detection of the A2143G, A2142C, and A2142G mutations. The primers and HybProbes (hybridization probes) were obtained from Sentromer Ltd. (Istanbul, Turkey) and TibMolbiol GmBH (Berlin, Germany), respectively. RT-PCR reactions were performed using a LightCycler FastStart DNA Master HybProbe (Roche Diagnostics, Mannheim, Germany) kit on a LightCycler480 II (Roche Diagnostics, Mannheim, Germany) system according to the manufacturer’s instructions. A total of 5 µL of template DNA was used in the reactions, and the final volume of the reaction was 20 µL. RT-PCR reactions included an initial denaturation at 95 °C for 10 min followed by 45 cycles of amplification (10 s at 95 °C; 15 s at 60 °C) (Monocolor HybProbe Red 640 detection format, single read). The melting curve was 1 min at 95 °C, 1 min at 40 °C, and continuous reading at 75 °C. A2143G, A2142C, and A2142G mutation analyses were performed using Tm Calling analysis in LightCycler 480 1.5 software. All PCR reactions were conducted with a clarithromycin-sensitive *H. pylori* ATCC 43504 strain as a negative control. For positive controls, three plasmids were reconstructed from the records of GenBank for the A2143G, A2142G, and A2142C mutations [18]. The GenBank accession numbers of these plasmids corresponding to mutations A2142C, A2142G, and A2143G are AF550407, AF550408, and AF550409, respectively.

### 2.5. Sequencing of the 23S rDNA

All of the RT-PCR products were sequenced. Each real-time PCR product was separated on 2% agarose gels, and amplification fragments were purified using a QIAquick PCR Purification Kit (Qiagen GmBH, Hilden, Germany) according to the manufacturer’s instructions. The amplification products were sequenced in two directions (forward and reverse primers) using a BigDye Terminator Cycle Sequencing kit (Applied BioSystems, Foster City, CA, USA) on an ABI PRISM 3130 Genetic Analyzer (Applied BioSystems, Foster City, CA, USA) according to the manufacturer’s instructions. The results were analyzed by Sequence Analysis Software (Applied Biosystems, Foster City, CA, USA). Then, the alignments of single consensus sequences were analyzed using the Clustal W menu in MEGA X software. FASTA files were compared with the reference sequence of the 23S rRNA gene (GenBank Accession number: U27270.1).

### 2.6. Statistical Analysis Methods

The chi-squared test and Fisher’s exact test for multiple comparisons were used to compare subgroups. The results are reported as the Benjamini–Hochberg-adjusted *p*-value. All the reported confidence intervals (CIs) were calculated at the 95% level. Significance was recognized when *p* < 0.05. We used SPSS 25.0 (IBM Corporation, Armonk, New York, NY, USA) for the analyses of the variables.

## 3. Results

The fecal *Helicobacter pylori* (HP) antigen ELISA test was positive in 18.6% (101/543) of the symptomatic (49) and asymptomatic (52) children tested at our pediatric outpatient clinic. They were selected randomly according to their symptoms. The mean age was 7.13 years for all groups: 2.26 years for the 0–5-year-old group and 11.37 years for the 5–18-year-old group. The characteristics of the different age groups are shown in Table 2(a,b).

A2142G mutation was detected in one (2%) and six (11.3%) of the 0–5-year-olds and 5–18-year-olds, respectively. No statistical differences were found between the 0–5-year-old and 5–18-year-old groups (*p* = 0.115). A2143G mutation was detected in 0 (0%) and 11 (20.7%) of the 0–5-year-olds and 5–18-year-old groups, respectively. The A2143G mutation was significantly more likely to be detected in the 5–18-year-old group than in the 0–5-year-old group (*p* = 0.001). The A2142C mutation was not detected in any of the H. pylori DNA isolates. Our overall positivity rate was 17.82% (18/101) for clarithromycin resistance via detecting A2142G and A2143G mutations (Table 3).

The A2143G mutation was detected in nine (18.4%) and two (3.8%) of the symptomatic and asymptomatic children, respectively. Symptomatic children were significantly more likely to test positive for A2143G compared with asymptomatic children (*p* = 0.019) (Table 4).

The A2142G mutation was detected in five (10.2%) and two (3.8%) of the symptomatic and asymptomatic children, respectively. There was no significant difference between the symptomatic and asymptomatic groups (*p* = 0.209) (Table 5).

The A2142G mutation was detected in one (3.8%) and none (0%) of the symptomatic and asymptomatic children in the 0–5-year-old group, respectively. The A2143G mutation was not detected in any of the children. No significant difference was detected between symptomatic and asymptomatic children in terms of A2142G and A2143G mutations in the 0–5-year-old group (Table 6).

The A2142G mutation was detected in four (17.4%) and two (6.7%) of the symptomatic and asymptomatic children in the 5–18-year-old group, respectively. No significant difference was detected between symptomatic and asymptomatic children in terms of the A2142G mutation in 5–18-year-olds (*p* = 0.383). The A2143G mutation was detected in nine (39.1%) and two (6.7%) of the symptomatic and asymptomatic children in the 5–18-year-old group, respectively. Symptomatic children were significantly more likely to have the A2143G mutation than asymptomatic children in the 5–18-year-old group (Table 7).

Seven (87.5%) of the eight A2142G-positive samples were cagA positive and one (12.5%) was not; 81 (87%) of the 90 samples without the A2142G mutation were positive for cagA. No significant difference was detected between cagA positivity and A2142G mutations (*p* = 0.974) (Table 8).

Eleven (91.7%) of the twelve samples with A2143G mutations were cagA positive, and one (8.3%) was cagA negative; 77 (86.5%) of the 90 samples without A2143G mutations were positive for cagA. No significant difference was detected between cagA positivity and A2143G mutations (*p* = 0.617) (Table 9).

No significant difference was detected between males and females in terms of either the A2142G or A2143G mutation. The point mutations were confirmed via sequencing.

## 4. Discussion

There is a growing interest in possible links between *H. pylori* infection and extra-gastric disorders. Although *H. pylori* is most frequently acquired during childhood in both developed and developing countries, this association is not considered in clinical practice. Thus, large and well-designed trials are needed among symptomatic and asymptomatic children living in areas with high and low rates of *H. pylori* infection [19]. Clarithromycin is commonly used in the respiratory tract and as the standard therapy in *H. pylori* infections for adults and children [20]. Clarithromycin binds to the 23S ribosomal RNA gene and inhibits protein synthesis. Point mutations at positions 2142 or 2143 cause resistance against clarithromycin. It is still debated whether to use antimicrobials for the treatment of children with *H. pylori* infection. According to the Maastricht IV Consensus and ESPGHAN/NASPGHAN guidelines, chronic idiopathic thrombocytopenic purpura, iron deficiency anemia, and peptic ulcer are the cases in which the eradication of *H. pylori* is recommended in children [5,6]. The difficult and time-consuming nature of traditional susceptibility testing has led researchers to search for non-invasive, fast, and reliable testing methods such as rapid genotypic susceptibility determination from the stool samples of children using molecular diagnostics [21].

We evaluated the clarithromycin resistance of *H. pylori* DNAs isolated from the stool samples of 101 children: 49 symptomatic children with abdominal pain, halitosis, and diarrhea and 52 asymptomatic children. We detected A2143G mutations in 10.9% and A2142G mutations in 6.9% of children. The most commonly detected mutation was the A2143G mutation in 5–18-year-olds (39.1%). The overall prevalence of clarithromycin resistance in *H. pylori* in our 101 symptomatic and asymptomatic children was determined to be 17.82% (18/101). Moreover, we detected A2143G mutations in 18.4% and 3.8% of the symptomatic and asymptomatic children, respectively, and found a significant difference between symptomatic and asymptomatic children in terms of A2143G mutations in favor of symptomatic children. However, there was no significant difference for A2142G mutations, and we did not find any A2142C mutations in our *H. pylori* DNA isolates. We also found a significant difference between the two age groups when it came to A2143G mutations, with more being present in the older group. A significant increase in A2143 mutations was also obvious in symptomatic children in the 5–18-year-old group. There was no association between *cagA* status and either A2143G or A2142G mutations.

In a very recent 2022 Turkish study, the *H. pylori* infection frequency in 804 children with abdominal pain was reported to be 22.3% [22]. In a study before 2000, *H. pylori* infection frequency was reported to be 52.5%, and this was probably a result of the excessive use of macrolide group antibiotics [23]. Prior studies in Turkey have found clarithromycin resistance in children with *H. pylori* infection; these rates varied between 18.2% and 25.7%, which is similar to our overall clarithromycin resistance rate (17.8%) [24,25]. In a recent study in Turkey, *H. pylori* resistance to clarithromycin was found in 27% of Turkish children aged 2–18 years who had a history of epigastric pain and/or nausea [26]. In another Turkish study, clarithromycin resistance was detected in 28 (30.1%) of 93 pediatric gastroenterology patients [27]. Furthermore, clarithromycin resistance was detected in 29 (25.7%) children between 3 and 18 years of age with *H. pylori* infection in another Turkish study [28]. Since very few studies have been carried out on this subject in Turkey, we suggest that this study will be very useful for raising awareness of pediatric *H. pylori* infections among pediatricians. The results of the limited number of studies that have been conducted also seem to be compatible with the results of our study. Point mutations vary in different populations and regions, and the A2143G mutation is a prominent mutation in clarithromycin-resistant *H. pylori* strains in Europe, as can be seen from our *H. pylori* DNA isolates [9]. The A2143G mutation is predominant in European and South American studies, but the A2144G mutation is predominant in Asia and Africa, except in China, where A2143G is predominant [9].

In one of the very limited studies evaluating point mutations conferring clarithromycin resistance to *H. pylori*, Booka et al. [29] (2015) studied 23 children aged between four months and 16 years without significant upper abdominal symptoms and found that 18 of them tested positive via the *Helicobacter pylori* stool antigen (HpSA) test. Four (25%) and one (6%) of the 16 PCR-positive samples were associated with A2143G and A2142G mutations, respectively. They detected a clarithromycin resistance rate of *H. pylori* of 31% (5/16). The prevalence of clarithromycin-resistant *H. pylori* strains in our study (17.82%) was lower than that found in Booka et al.’s study (31%). A retrospective cohort study reported that certain point mutations such as A2143G mutations may result from clarithromycin-based treatment failure [30]. Therefore, the detection of mutations related to the 23S rRNA gene needs to be conducted before treatment, especially in regions with clarithromycin resistance rates over 15–20% [5].

To give a broader perspective, A2143G and A2142G mutations confer clarithromycin resistance in 95% of cases in Japan [31]; A2143G, A2142G, and A2142C are responsible for 97.7% of cases in Europe [13]; and A2143G and A2142G are responsible for 91.4% of cases in the United States [32]. These studies showed that clarithromycin resistance can be determined by the detection of these mutations. Our results are consistent with European and Japanese studies showing that mutation A2143G is the most frequent among clarithromycin-resistant *H. pylori* DNA isolates [10,13,31]. Beer-Davidson et al. [33] studied the stool samples of 188 healthy children aged 6–9 years and 272 samples of 92 infants aged 2–18 months; they determined the rates of clarithromycin resistance and *cagA* gene positivity in *H. pylori*-positive samples. They found that 16 samples were positive for the *cagA* gene and that three were positive for the mutation (A2143G) via multiplex real-time polymerase chain reaction (q-PCR). This suggested that the use of clarithromycin in the community may induce mutations in *H. pylori* DNA based on selective pressure from antibiotics. Children may have acquired an *H. pylori*-resistant strain from their family members. To confirm this hypothesis, the intrafamilial transmission of *H. pylori* was documented in a previous report [34]. Their A2143G mutation rate was lower than that found in our results. Moreover, Serrano et al. [35] detected point mutations in the 23S-rRNA gene in 53 *H. pylori* isolates from biopsy samples. They detected 11 (21%) point mutations in the 23S-rRNA gene and determined that the A2143G mutation was the most frequent mutation in ten isolates from children (19%); one (2%) had the A2142G mutation. Their results agree with ours.

In another Turkish study, Çagan-Appak et al. [36] reported resistance to clarithromycin in gastric biopsies of 19/200 (9.5%) children with dyspepsia and abdominal pain; 10/19 (52.6%) had A2143G mutations, and 9/19 (47.4%) had A2142G mutations. There were no A2142C mutations. They detected no significant relationship between upper gastrointestinal (GI) system indications and the detection of clarithromycin resistance. Their A2143G and A2142G mutation rates were higher than ours, but this may be the result of their cases all having dyspepsia and abdominal pain. The A2142C mutation was not detected in any children in this study. A2143G mutations were the most frequent. Several other studies have results that are consistent with ours [30,37,38,39,40]. There is an approximately 50% treatment failure rate, but this ratio is low for A2142G and A2142C mutations (20% of children) [30].

Taneika et al. [41] suggested that *cagA*-positive *H. pylori* strains seem to be more easily eradicated by antibiotics, perhaps due to the fast-growing nature of *cagA*-positive strains. *cagA*-negative strains may easily acquire drug resistance because of the selection pressure of antibiotics. *cagA*-negative *H. pylori* strains cause less severe inflammation, and this makes them less accessible to antibiotics. Therefore, it is difficult to eradicate *cagA*-negative *H. pylori*. Importantly, we could not find any association between *cagA* positivity and point mutations conferring clarithromycin resistance.

The molecular characterization of infectious *H. pylori* strains such as the *cagA* virulence gene and clarithromycin resistance mutations are not possible if only EIA is used. Another advantage of investigating clarithromycin resistance in stool samples is that fecal material can be obtained easily [42]. It is very important to know the susceptibility status of *H. pylori* strains to clarithromycin before starting treatment—especially for children. In other words, the rapid detection of clarithromycin resistance is necessary for effective antibiotic therapy and to monitor resistance for diseases such as chronic idiopathic thrombocytopenic purpura, iron deficiency anemia, and peptic ulcer in children. Traditional susceptibility testing takes at least 10 to 14 days (e.g., the E-test). There may be different results between phenotypical and genotypical clarithromycin susceptibility tests because genotypical methods only detect A2142G, A2143G, and A2142C mutations, but there are additional rare strains [43]. If *H. pylori* eradication fails, then A2143G and A2142G mutations indicate that clarithromycin resistance should be considered the primary cause. Clarithromycin resistance rates have been reported to be between 13.9% to 84.9% for children in 10 countries [11]. Moreover, the rate of clarithromycin resistance has increased in Japan, Austria, Bulgaria, and South Korea. The type of mutation is not the same for every country. The A2144G mutation is most common in clarithromycin-resistant *H. pylori* strains in South Korea and Japan [4]. Clarithromycin resistance is likely caused by the use of macrolides in pediatric, respiratory, and otorhinolaryngology infections [44].

The “test-and-treat” strategy is not recommended for the treatment of *H. pylori* infection because the primary purpose of testing is to understand the cause of the clinical symptoms. Since it has been proven that *H. pylori* infection is not associated with specific symptoms, this strategy cannot provide this information in children [45]. Moreover, gastro endoscopy is an invasive procedure and is difficult to conduct; it is not recommended for children. The detection of point mutations in the 23S rRNA of *H. pylori* is a very promising non-invasive method, especially for children. Agar dilution or the E-test are culture-based methods used to determine clarithromycin resistance, but they are slow and expensive and require a special lab [12.33].

Clarithromycin resistance mainly results from specific mutations, and thus, molecular methods can quickly give reliable results. A strong association between clarithromycin resistance and A2142G, A2143G, and A2142C mutations in the 23S rDNA has previously been documented. RT-PCR is a fast and reliable detection method for the detection of clarithromycin resistance from stool samples. Clinicians quickly learn the status of clarithromycin resistance and prescribe proper antibiotic therapy if no clarithromycin resistance is detected. These genotypical methods for the detection of clarithromycin resistance are especially important for unsuccessful *H. pylori* cultures [46]. Determining clarithromycin resistance by detecting point mutations in the 23S rRNA of *H. pylori* has some advantages such as high specificity, rapidity, and independence of bacteria growth, as well as good reproducibility and easy standardization. These techniques detect both the presence of *H. pylori* and clarithromycin resistance at the same time [9].

Rimbara et al. [47] compared the detection of clarithromycin resistance from feces and gastric juice via agar dilution and nested PCR. They found that nested PCR was more sensitive than agar dilution in the detection of clarithromycin resistance. In Austria, a stool real-time PCR was compared with culture-based clarithromycin susceptibility testing for the detection of *H. pylori* infection and clarithromycin resistance. They suggested that stool PCR was as effective as invasive methods during the treatment of children (stool PCR group 78.2% vs. gastric biopsy group 73.2%, *p* = 0.63); they concluded that this method can be used to monitor the secondary clarithromycin resistance following eradication treatment [48].

As a limitation of this study, it was not possible to compare our results with the results obtained with standard methods because we did not have gastric biopsy samples of the children. In a very similar study, Osaki et al. [12] compared the results of clarithromycin resistance-associated mutations using DNAs obtained from feces with conventional methods, and they suggested that this method was considered useful as an alternative method for the detection of antibiotic-resistant *H. pylori* infections in children. They concluded that, even though the specificity was lower than that of conventional methods, this fecal detection method can be used for asymptomatic individuals, including children. There may also be mixed clarithromycin-sensitive and -resistant *H. pylori* strains in the same person. Noguchi et al. [14] and Osaki et al. [12] reported on this detection method for the detection of clarithromycin resistance, and it was shown to be highly sensitive but to have lower specificity than conventional methods, including culture and susceptibility testing using cultured isolates, and these false negatives may lead to eradication failure.

Although it was reported that no clinical manifestations, especially not recurrent abdominal pain, have been shown to be specific to *H. pylori* infection in children, this study reflects the high levels of clarithromycin resistance in *H. pylori* samples isolated from the stools of symptomatic and, to a lesser extent, asymptomatic children [49]. We detected a significant difference between symptomatic and asymptomatic children in terms of A2143G mutations in the 5–18-year-old group. The abundance of point mutations in this group makes us think that *H. pylori* strains in this group acquired a persistent character and are more likely to be exposed to the selective pressure effect of antibiotics. Macrolides are frequently used in upper respiratory tract infections in children from this age group. Alternatively, these clarithromycin-resistant *H. pylori* strains may be acquired from family members, or children may be infected by environmental sources. We may suggest that *H. pylori* strains carrying mutations in the 23S RNA subunit conferring clarithromycin resistance may develop an intense inflammatory response in the gastric epithelial cells and, thus, can proliferate more rapidly, possibly causing diarrhea, halitosis, or abdominal pain in the 5–18-year-old group. It was reported that most people with mild chronic inflammation have *H. pylori* strains without mutations, and most people with moderate to severe inflammation have mutant (A2142G and A2143G point mutations) *H. pylori* strains in a very recent 2021 study [50], which supports our hypothesis. The low sensitivity of stool PCR for detecting *H. pylori* infection compared with HpSA in children was previously solved via HpSA to detect *H. pylori*-positive samples. Pediatricians should consider *H. pylori* infection, especially in symptomatic children more than five years old, and evaluate both the presence and point mutations conferring clarithromycin-resistant *H. pylori* strains in a short time using non-invasive methods such as real-time PCR. This may help to avoid treatment failure if clarithromycin resistance is present.

Endoscopic examination for *H. pylori* infections in children is not common, and therefore, it is not possible to carry out assays for a conventional method using isolated *H. pylori* strains for the detection of antibiotic resistance. Therefore, we suggest that this PCR-based molecular method using fecal DNA can be considered as an alternative method for the detection of resistance in *H. pylori* infections. It may be useful to use this method for the detection of point mutations conferring resistance from the clinical point of view, but this method should be used with conventional culture methods. When the screening and testing of *H. pylori* infection in children is not possible, this molecular method seems to be useful for *H. pylori* resistance detection. Considering the limited pediatric data, new and larger-scale studies are needed on clarithromycin-resistant *H. pylori* in children from different regions of the world.

## Figures and Tables

**Table 1 diagnostics-12-02119-t001:** The hybridization probes and primers used for the detection of the *cagA* gene and A2143G, A2142C, and A2142G mutations.

Primers/Probe Name	Oligonucleotides	Nucleotide Positions	PCR Conditions	Ref.
** *cagA* ** **Forward** ***cagA* Reverse**	5-GAT AAC AGG CAA GCT TTT GAG G CTG-3 5-CAA AAG ATT GTT TGG CAG A-3		1 cycle at 95 °C for 2 min, 45 cycles of 95 °C for 30 s, 45 s at 53 °C, and 45 s at 72 °C, 1 cycle 5 min at 72 °C	[16]
**HPY-S**	5-AGGTTAAGAGGATGCGT CAGTC-3	1931–1952	1 cycle at 95 °C for 10 min, 45 cycles of 10 seconds at 95 °C and 15 seconds at 60 °C, 1 cycle for 1 minute at 95 °C, 1 minute at 40 °C, and continuous reading at 75 °C	[17]
**HPY-A**	5-CGCATGATATTCCCATTAGC AGT-3	2175–2197		[17]
**Red640**	5-GGCAAGACGGAAAGACC-3	2504–2520		[18]
**5Flour**	5-TGTAGTGGAGGTGAAAATTCCTCCTACCC-3	2473–2501		[18]

**Table 2 diagnostics-12-02119-t002:** (**a**). The characteristics of the different age groups. (**b**) *H. pylori* positivity of different age groups.

**(a)**					
**Age Group**	** *n* ** **(%)**	**Male/Female**	**Symptomatic (Abdominal Pain, Halitosis, and Diarrhea)**	**Asymptomatic**	**cagA Positivity**
**0–5 years**	48 (46.53%)	22/26	26 (54.2%)	22 (45.8%)	44
**5–18 years**	53 (53.47%)	27/26	23 (43.4%)	30 (56.65%)	43
**Total**	101		49	52	87
**(b)**					
**Age Group**	0–2 Years (Including 2 Years) (*n* (%))	2–5 Years (Including 2 Years) (*n* (%))	5–10 Years (Including 10 Years) (*n* (%))	10–14 Years (Including 14 Years) (*n* (%))	14–18 Years (*n* (%))
** *H. pylori* ** **positivity**	30 (29.7%)	18 (17.8%)	23 (22.8%)	16 (15.8%)	14 (13.9%)
**Symptomatic/Asymptomatic**	17/13	9/9	8/15	9/7	6/8

**Table 3 diagnostics-12-02119-t003:** The distribution of A2142G and A2143G mutations by age group.

Age Group	A2142G		*p*	A2143G		p
	+	−		+	−	
**0–5 years**	1 (2%)	47 (98%)	**0.115**	0 (0%)	48 (100%)	**0.001**
**5–18 years**	6 (11.3%)	47 (88.7%)		11 (20.7)	42 (79.3%)	
**Total**	7	94		11	90	

**Table 4 diagnostics-12-02119-t004:** The distribution of the A2143G mutation by symptom group.

Symptom Group	A2143G		*p*
	+	−	
**Symptomatic**	9 (18.4%)	40 (81.6%)	**0.019**
**Asymptomatic**	2 (3.8%)	50 (96.2%)	
**Total**	11	90	

**Table 5 diagnostics-12-02119-t005:** The distribution of A2142G mutation by symptom group.

Symptom Group	A2142G		*p*
	+	−	
**Symptomatic**	5 (10.2%)	44 (89.8%)	**0.209**
**Asymptomatic**	2 (3.8%)	50 (96.2%)	
**Total**	7	94	

**Table 6 diagnostics-12-02119-t006:** The distribution of A2142G and A2143G mutations by symptom group in the 0–5-year-olds.

Mutations by Symptom Group in the 0–5-Year-Olds	A2142G (*n* (%))		*p*	A2143G (n (%))		*p*
	+	−		+	−	
**Symptomatic**	1 (3.8%)	25 (96.2%)	**1**	0 (0%)	26 (100%)	**-**
**Asymptomatic**	0 (0%)	22 (100%)		0 (0%)	22 (100%)	
**Total**	1	47		0	48	

**Table 7 diagnostics-12-02119-t007:** The distribution of A2142G and A2143G mutations by symptom group in the 5–18-year-olds.

Mutations by Symptom Group in the 5–18-Year-Olds	A2142G		*p*	A2143G		*p*
	+	−		+	−	
**Symptomatic**	4 (17.4%)	19 (82.6%)	0.383	9 (39.1%)	14 (60.9%)	0.006
**Asymptomatic**	2 (6.7%)	28 (93.3%)		2 (6.7%)	28 (93.3%)	
**Total**	6	47		11	42	

**Table 8 diagnostics-12-02119-t008:** The distribution of A2142G mutation by cagA status in all age groups.

	cagA + (*n*: 88)	cagA − (*n*:13)	Total	*p*	OR	95% C.I. Lower–Upper
**A2142G +** **(*n*:11)**	7 (87.5%)	1 (12.5%)	8	0.974	1.037	0.117–9.185
**A2142G −** **(*n*:90)**	81 (87%)	12 (13%)	93			
**Total**	88	13	101			

**Table 9 diagnostics-12-02119-t009:** The distribution of A2143G mutation by cagA status in all age groups.

	cagA + (*n*=88)	cagA − (*n*=13)	Total	*p*	OR	95% C.I. Lower–Upper
**A2143G +** **(*n*:11)**	11 (91.7%)	1 (8.3%)	12	0.617	1.714	0.203–14.506
**A2143G −** **(*n*:90)**	77 (86.5%)	12 (13.5%)	89			
**Total**	88	13	101			

## Data Availability

The datasets generated for this study can be found as the 23s sequencing data in GenBank at https://www.ncbi.nlm.nih.gov/nuccore/ON688309 (accessed on 2 May 2022).

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
