# Peer review of "Detection of A2143G, A2142C, and A2142G Point Mutations with Real-Time PCR in Stool Specimens from Children Infected with Helicobacter pylori"

_diagnostics, 2022, doi:10.3390/diagnostics12092119_

Round 1
Reviewer 1 Report
I had the pleasure to review the work "Detection of A2143G, A2142C, and A2142G point mutations by real-time PCR in stool specimens from children infected with Helicobacter pylori". Looking at the approach to mutation conditioning clarithromycin resistance, we know that the work is not innovative. Here, the source of research material is important, allowing the omission of difficult-to-obtain gastric biopsies, especially in children, with use of biopsy. The work is written well. Here, a good and comprehensive discussion deserves recognition. After major corrections, I believe the work should be published in Diagnostics.
Major revision:
1) In the title and abstract, please use italics for the name Helicobacter pylori.
2) Tables - please rewrite the tables so as not to separate individual words or reaction stages. This makes the data unreadable. Maybe reduce the font ?
3) Materials and methods - probes were used for the reaction - then why was a melting curve made ?
4) Materials and methods - please indicate which version of the CLUSTAL program was used for the analysis.
5) Materials and methods - there is no subchapter on the applied statistical tests and software to perform statistics.
6) I have doubts about the reliability of the obtained results. Since it is an innovative approach to such diagnostics, the results should be compared with the results obtained with standard methods. Please complete the discussion with this thread.
Author Response
Major revision:
1) In the title and abstract, please use italics for the name Helicobacter pylori.
- Helicobacter pylori was revised as italic type both in the title and abstract.
2) Tables - please rewrite the tables so as not to separate individual words or reaction stages. This makes the data unreadable. Maybe reduce the font ?
- Font and paragraphs were reduced.
3) Materials and methods - probes were used for the reaction - then why was a melting curve made ?
- This is a hybridization probe, not a taqman hydrolysis probe. Melting curve cannot be made in hydrolysis probe but hybridization probe is a probe format developed by Roche and snp analysis can be done by performing melting curve analysis in this format. You can look the technical note by thi link: https://biochimie.umontreal.ca/wp-content/uploads/sites/37/2015/11/assay_formats.pdf
4) Materials and methods - please indicate which version of the CLUSTAL program was used for the analysis.
- The following sentence was added to the Material and Methods section “Then, alignment of single consensus sequences were analyzed using Clustal W menu with MEGA X software.”.
5) Materials and methods - there is no subchapter on the applied statistical tests and software to perform statistics.
- The folloing subchapter was added at the end of Materials and Methods Section.
Statistical Analysis Methods
The chi-squared test and Fisher’s exact test for multiple comparisons were used to com-pare subgroups (Tables 3–9). The results are reported as the Benjamini–Hochberg-adjusted P-value. All the reported confidence intervals (CIs) were calculated at the 95% level. Significance was recognized when P<0.05. We used SPSS 25.0 (IBM Corpo-ration, Armonk, New York, USA) for the analyses of the variables
6) I have doubts about the reliability of the obtained results. Since it is an innovative approach to such diagnostics, the results should be compared with the results obtained with standard methods. Please complete the discussion with this thread.
- We agree with the reviewer and added the following new paragraph to the Discussion “As a limitation of this study, it was not possible to compare our results with the results obtained with standard methods because we did not have gastric biopsy samples of the children. In a very similar study, Osaki et al. [12] compared the results of clarithromycin resistance-associated mutations using DNAs obtained from feces with conventional methods, and they suggested that this method was considered useful as an alternative method for the detection of antibiotic-resistant H. pylori infections in children. They con-cluded that even though the specificity was lower than that of conventional methods, this fecal detection method can be used for asymptomatic individuals, including children. There may also be mixed clarithromycin sensitive and resistant H. pylori strains in the same person. Noguchi et al. [14] and Osaki et al. [12] reported on this detection method for the detection of clarithromycin resistance, and it was shown to be highly sensitive but to have lower specificity than conventional methods, including culture and susceptibility testing using cultured isolates, and these false negatives may lead to eradication failure.”.

Reviewer 2 Report
1-Abbreviations should be correctly stated in the abstract and introduction
2-The text of the article should be written in native English without grammatical errors
3-Are the primers checked with primer blast or oligo to check the quality after selection?
4-Was the disease of all the children confirmed by endoscopy, and what conditions were used to select the groups and confirm the disease and the history of the people's disease? And did the people have other underlying diseases and what was the grade of their stomach infection and disease?
5-Please, after discussing the conclusion and innovation, the work and service that can be done to humanity from this article should be clearly stated.
6-Please explain the discussion in a more concise and expressive way for the general readers?
Author Response
Answers To the Comments
1-Abbreviations should be correctly stated in the abstract and introduction
- In the Abstract, following abbreviations “23S ribosomal RNA (rRNA), Helicobacter pylori (H. pylori), enzyme linked ımmunosorbent assay (ELISA), real-time polymerase chain reaction (RT-PCR)” were correctly stated. In the Introduction “mucosa-associated lymphoid tissue (MALT), polymerase chain reaction (PCR), The cytotoxin-associated antigen A (CagA), real-time polymerase chain reaction (RT-PCR),” were also correctly stated.
2-The text of the article should be written in native English without grammatical errors
2- The manuscript was already edited by Amerian Manucript Editors but after revisions I also sent it to MDPI editing service for re-editing (English-Editing-Certificate-49546 by MDPI). I will also upload the certificate to the system.
3-Are the primers checked with primer blast or oligo to check the quality after selection?
- Primer probe design controls were not made because the primers were taken from the similar studies (references) made before.
4-Was the disease of all the children confirmed by endoscopy, and what conditions were used to select the groups and confirm the disease and the history of the people's disease? And did the people have other underlying diseases and what was the grade of their stomach infection and disease?
- The disease of all the children was not confirmed by endoscopy. The conditions that were used to select the groups and confirm the disease are; As we indicated in the Methods section, we selected the first 3 children who applied to the pediatric outpatient clinic on working days were asked to give stool samples regardless of their clinical characteristics. The children were screened regardless of their complaints, and stool antigen test, which is an indicator of active infection, was performed. In the frequency studies conducted in the pediatric age group in our country, mostly serological methods and ELISA methods were used. Antibody IgG test against H. pylori by ELISA method may result in positive results for a long time. It has been reported that serological tests, which were advantageous in terms of easy applicability and cheapness in the past, can continue to be positive for 3 years despite treatment. For this reason, it is not reliable for diagnosis or treatment. For the history of the people's disease, The parents of the subjects were asked to complete a questionnaire. The children don’t hve any other underlying diseases. The children who were admitted to the general pediatry clinic of our university hospital and having no underlying disease were selected The most common symtom of the H pylori positive children was diarrhea (82.4%). No other symptoms were observed except abdominal pain, halitosis and diarrhea in the H pylori positive children. In fact, most of the researcher suggested that no clinical manifestations have been shown to be specific to H. pylori infection in children (a).
- Kotilea K, Kalach N, Homan M, Bontems P. Helicobacter pylori infectionin pediatric patients: update on diagnosis and eradication strategies. Paediatr Drugs. 2018;20(4):337–51.
5-Please, after discussing the conclusion and innovation, the work and service that can be done to humanity from this article should be clearly stated.
- The following paragraph was added at the end of Discussion. “Endoscopic examination for H. pylori infections in children is not commonly an option chosen and therefore it is not possible to carry out assays for a conventional method using isolated H. pylori strains for the detection of antibiotic resistance. Therefore, we suggest that this PCR-based molecular method using fecal DNA can be considered as an alterna-tive method for the detection of resistance in H. pylori infections. it may be suggested use-ful to use this method for the detection of point mutations conferring resistance from the clinical point of view, but this method should be used with conventional culture methods. When the screening and testing of H. pylori infection in children is not possible, this mo-lecular method seems to be is useful for H. pylori resistance detection. Considering the limited pediatric data, new and larger-scale studies are needed on clarithromy-cin-resistant H. pylori in children from different regions of the world. “
6-Please explain the discussion in a more concise and expressive way for the general readers?
- I suggest the following new paragraph and the answer for 5. Comment are enough to understand in a more concise and expressive way for the general readers.
” As a limitation of this study, it was not possible to compare our results with the results obtained with standard methods because we did not have gastric biopsy samples of the children. In a very similar study, Osaki et al. [12] compared the results of clarithromycin resistance-associated mutations using DNAs obtained from feces with conventional methods, and they suggested that this method was considered useful as an alternative method for the detection of antibiotic-resistant H. pylori infections in children. They concluded that even though the specificity was lower than that of conventional methods, this fecal detection method can be used for asymptomatic individuals, including children. There may also be mixed clarithromycin sensitive and resistant H. pylori strains in the same person. Noguchi et al. [14] and Osaki et al. [12] reported on this detection method for the detection of clarithromycin resistance, and it was shown to be highly sensitive but to have lower specificity than conventional methods, including culture and susceptibility testing using cultured isolates, and these false negatives may lead to eradication failure.”
“Endoscopic examination for H. pylori infections in children is not common, and therefore it is not possible to carry out assays for a conventional method using isolated H. pylori strains for the detection of antibiotic resistance. Therefore, we suggest that this PCR-based molecular method using fecal DNA can be considered as an alternative method for the detection of resistance in H. pylori infections. It may be useful to use this method for the detection of point mutations conferring resistance from the clinical point of view, but this method should be used with conventional culture methods. When the screening and test-ing of H. pylori infection in children is not possible, this molecular method seems to be useful for H. pylori resistance detection. Considering the limited pediatric data, new and larger-scale studies are needed on clarithromycin-resistant H. pylori in children from dif-ferent regions of the world.”

Round 2
Reviewer 1 Report
Thank you very much. The authors answered all my questions satisfactorily and revised the manuscript according to my suggestions.
Reviewer 2 Report
accept